# Deep Graph-Level Clustering Using Pseudo-Label-Guided Mutual Information Maximization Network

## Abstract

In this work, we study the problem of partitioning a set of graphs into different groups such that the graphs in the same group are similar while the graphs in different groups are dissimilar. This problem was rarely studied previously, although there have been a lot of work on node clustering and graph classification. The problem is challenging because it is difficult to measure the similarity or distance between graphs. One feasible approach is using graph kernels to compute a similarity matrix for the graphs and then performing spectral clustering, but the effectiveness of existing graph kernels in measuring the similarity between graphs is very limited. To solve the problem, we propose a novel method called Deep Graph-Level Clustering (DGLC). DGLC utilizes a graph isomorphism network to learn graph-level representations by maximizing the mutual information between the representations of entire graphs and substructures, under the regularization of a clustering module that ensures discriminative representations via pseudo labels. DGLC achieves graph-level representation learning and graph-level clustering in an end-to-end manner. The experimental results on six benchmark datasets of graphs show that our DGLC has state-of-the-art performance in comparison to many baselines.

## 1 Introduction

Graph-structured data widely exist in real-world scenarios, such as social networks (Newman, 2006) and molecular analysis (Gilmer et al., 2017). Compared to other data formats, graph data explicitly contain connections between data through the attributes of nodes and edges, which can provide rich structural information for many applications. In recent years, machine learning on graph-structured data gains more and more attention. Many supervised and unsupervised learning methods have been proposed for graph-structured data in various applications.

The machine learning problems of graph-structured data can be organized into two categories: node-level learning and graph-level learning. In node-level learning, the samples are the nodes in a single graph. Node-level learning mainly includes node classification (Li et al., 2017; Wu et al., 2021; Xu et al., 2021) and node clustering (Wang et al., 2017; Pan & Kang, 2021; Lin et al., 2021). Classical node classification methods are often based on graph embedding (Yan et al., 2006; Cai et al., 2018) and graph regularization (Subramanya & Bilmes, 2009; Bhagat et al., 2011), while recent advances are based on graph neural networks (GNN) (Kipf & Welling, 2017; Xu et al., 2019; Wu et al., 2020). Owing to the success of GNN in nodes classification, a few researchers have proposed GNN-based methods for nodes clustering (Wang et al., 2019; Bo et al., 2020; Zhu & Koniusz, 2021).

Different from node-level learning, in graph-level learning, the samples are a set of graphs that can be organized into different groups. Classical methods for graph-level classification are often based on graph kernels (Vishwanathan et al., 2010; Yanardag & Vishwanathan, 2015) while recent advances are based on GNN (Wu et al., 2020; Rong et al., 2020). Researchers generally utilize various types of GNN, e.g., graph convolutional networks (GCNs) (Kipf & Welling, 2017) and graph isomorphism network (GIN) (Xu et al., 2019) to learn graph-level representations by aggregating inherent node information and structural neighbor information in graphs, then they train a classifier based on the learned graph-level representations (Zhang et al., 2018; Sun et al., 2020; Wang et al., 2021; Doshi

& Chepuri, 2022). Nevertheless, collecting large amounts of labels for graph-level classification is costly in real-world, and the clustering on graph-level data is much more difficult than that on nodes and still remains an open issue. It thereby shows the importance of exploring graph-level clustering, namely ***partitioning a set of graphs into different groups such that the graphs in the same group are similar while the graphs in different groups are dissimilar***.

Previous research on graph-level clustering is very limited. The major reason is that it is difficult to represent graphs as feature vectors or quantify the similarity between graphs in an unsupervised manner. An intuitive approach to graph-level clustering is to perform spectral clustering (Ng et al., 2001) over the similarity matrix produced by a graph kernels (Kondor & Pan, 2016; Du et al., 2019; Togninalli et al., 2019) on graphs. Although there have been a few graph kernels such as random walk kernel (Gärtner et al., 2003) and Weisfeiler-Lehman kernel (Shervashidze et al., 2011), most of them rely on manual design that fails to provide desirable generalization capability for various types of graphs and produce satisfactory similarity matrices for spectral clustering, which will be demonstrated in Section 4.3.

Another solution comes with the encouraging development of GNNs. Some latest works such as GCNs (Kipf & Welling, 2017) and GIN (Xu et al., 2019) have been proven to be effective in learning node/graph-level representations for various downstream tasks, e.g., node clustering (Wang et al., 2017; Bo et al., 2020; Liu et al., 2022) and graph classification (Sun et al., 2020; Sato et al., 2021; You et al., 2021)—thanks to the powerful generalization and representation learning capability of deep neural networks. Therefore, it may be possible to achieve graph-level clustering by performing classical clustering algorithms such as $k$-means (Hartigan & Wong, 1979) and spectral clustering over the graph-level representations produced by various unsupervised graph representation learning methods (Grover & Leskovec, 2016; Narayanan et al., 2017; Adhikari et al., 2018; Sun et al., 2020).

Although the afore-mentioned GNN-based unsupervised graph-level representation learning methods have shown promising performance in terms of some down-stream tasks such as node clustering and graph classification, they do not guarantee to generate effective features for the clustering tasks on entire graphs. In contrast, the graph-level clustering may benefit from an end-to-end framework that can learn clustering-oriented features in the graph-level representation learning. We summarize our motivation here: 1) Graph-level clustering is an important problem but it is rarely studied, though there have been a lot of works on graph-level classification and node-level clustering. 2) The performance of graph-kernels followed by spectral clustering and two-stage methods (deep graph-level feature learning followed by k-means or spectral clustering) haven't been well explored. 3) An end-to-end deep learning based graph-level clustering method is expected to outperform graph kernels and the two-stage methods because the feature learning is clustering-oriented. Therefore, we propose a novel graph clustering method called deep graph-level clustering (DGLC) in this paper. The proposed method is a fully unsupervised framework and yields the clustering-oriented graph-level representations via jointly optimizing two objectives: representation learning and clustering. The main contributions of this paper are summarized as follows.

- We investigate the effectiveness of various graph kernels as well as unsupervised graph representation learning methods in the problem of graph-level clustering.

- We propose an end-to-end graph-level clustering method. In the method, the clustering objective can guide the representation learning for entire graphs, which is demonstrated to be much more effective than those two-stage models in this paper.

- We conduct extensive comparative experiments of graph-level clustering on six benchmark datasets. Our method is compared with five graph kernel methods and four cutting-edge GNN representation learning methods, under the evaluation of three quantitative metrics and one qualitative (visualization) metric. Our method has state-of-the-art performance.

## 2 PRELIMINARIES

The notations used in this paper are shown in Table 1. In the next two subsections, we briefly introduce graph kernels and GNN based graph-level representation learning methods. We will also illustrate how to apply them to graph-level clustering.

Table 1: Notations for the main variables and parameters in this paper.

| | | | |
|---|---|---|---|
| $\mathcal{G}$ | Graph set | $\bar{G}$ | Graph set in a minibatch |
| $V$ | Node set | $E$ | Edge set |
| $\mathcal{X}$ | Node features set | $\mathcal{N}(v)$ | Neighborhood set of node $v$ |
| $G$ | A single graph | $K$ | Number of GNN hidden layers |
| $\mathbf{h}_v^k$ | Learned feature for node $v$ in $k$-th GNN layer | $\mathbf{a}_v^k$ | Aggregated feature for node $v$ in $k$-th GNN layer |
| $\mathbf{H}_\phi(G)$ | Graph-level representaion | $I_{\phi,\psi}$ | Mutual information estimator |
| $f_\theta$ | Cluster projector | $\mathbf{Z}_{\phi,\theta}(G)$ | Cluster embedding |
| $c$ | Number of clusters | $\phi$ | Parameters of GNN |
| $\psi$ | Parameters of mutual information estimator | $\theta$ | Parameter of clustering network |

## 2.1 GRAPH KERNELS

Graph kernels are techniques typically used in both supervised and unsupervised learning that exploit graph topology. They aim to learn graph representation implicitly with predetermined graph sub-structures. For a graph $G$, after its sub-graphs $\{G_i\}$ are defined, the kernel is calculated according to the occurrences of the sub-graphs of $\{G_i\}$. Namely, $\mathcal{K}_g(G_m, G_n) := \mathcal{F}_{G_m}^\top \mathcal{F}_{G_n}$, where $\mathcal{F}_{G_i}$ denotes frequency. In recent years, much effort has been devoted to the identification of desirable sub-graphs ranging from Graphlet kernel (Shervashidze et al., 2009), Random walk kernel (Vishwanathan et al., 2010), Shortest path kernel (Borgwardt & Kriegel, 2005) to Subgraph matching kernel (Kriege & Mutzel, 2012), Pyramid match kernel (Nikolentzos et al., 2017), etc. For example, one of the most popular kernels is the Weisfeiler-Lehman kernel (Shervashidze et al., 2011). It belongs to subtree kernel family and could scale up to large and labeled graphs. Weisfeiler-Lehman kernel is built upon other base kernels through Weisfeiler-Lehman test of isomorphism on graphs. The essential idea of Weisfeiler-Lehman kernel is to relabel the graph with not only the original label of each vertex, but also the sorted set of labels of its neighbors (sub-tree structure). With runtime scaling only linearly in the number of edges of the graphs, Weisfeiler-Lehman kernel is widely applied in computational biology and social network analysis. However, Weisfeiler–Lehman kernel's hashing step is somewhat ad-hoc, with performance varying from data to data (Kondor & Pan, 2016). Another state-of-the-art algorithm is the shortest-path kernel (Borgwardt & Kriegel, 2005), which is based on paths instead of conventional walks and cycles. By transforming the original graph into shortest-paths graph $\tilde{G}_{v,u,e} = \{\text{the number of occurrences of vertex } v \text{ and } u \text{ connected by shortest-path } e\}$, it avoids the high computational complexity of graph kernels based on walks, subtrees and cycles. In this paper, several graph kernels are selected as comparative models to test their efficiency on clustering. More specifically, we perform spectral clustering with the similarity matrices computed by graph kernels. One limitation is that existing graph kernels are not effective enough to quantify the similarity between graphs. In addition, most of them cannot take advantages of the nodes features and labels of graph. The related results and time complexity comparison can be found in Table 3-5 and Appendix A.5.

## 2.2 UNSUPERVISED GRAPH-LEVEL REPRESENTATION LEARNING

In recent years, GNN related models (Wu et al., 2020; Zhou et al., 2020) have shown state-of-the-art performance in many graph-data related tasks such as nodes classification (Kipf & Welling, 2017; Zhang et al., 2019) and graph classification (Zhang et al., 2018; Xu et al., 2019; Sun et al., 2020). A number of graph representation learning methods have been proposed to handle the graph/node classification and node clustering tasks. For example, Grover & Leskovec (2016) proposed to learn low-dimensional mapping for nodes that maximally preserves the neighborhood information of nodes. Veličković et al. (2019) proposed to learn node representations for node classification via maximizing the mutual information between the patch representations and summarized graph representations. Similarly, Sun et al. (2020) utilized the mutual information maximization strategy and GIN (Xu et al., 2019) to learn graph representations for graph-level classifications. You et al. (2020; 2021) took inspiration from the self-supervised learning to augment the graph data to construct positive/negative pairs, thereby learn effective graph representations with contrastive learning strategy (Chen et al., 2020).

It should be pointed out that existing graph representation learning methods rarely investigate the graph-level clustering task, as it is far difficult than graph classification or node clustering. An intuitive strategy is to perform $k$-means (Hartigan & Wong, 1979) or spectral clustering (Ng et al., 2001)

on the learned graph-level representations given by those methods. Nevertheless, the clustering performance is not desirable as can be observed in Section 4.4, because the representations learned by those methods are not guaranteed to be suitable or effective for graph-level clustering. Therefore, we present our DGLC method to investigate the way to learn clustering-oriented graph-level representations, of which the learning is guided by an explicit clustering objective.

## 3 METHODOLOGY

### 3.1 PROBLEM FORMULATION

Given a set of $n$ graphs, i.e., $\mathcal{G} := \{G_1, G_2, \ldots, G_n\}$, where the $i$-th graph $G_i = (V_i, E_i)$ has node features $\mathbf{X}_i = \{\mathbf{x}_v^{(i)}\}_{v \in V_i}$ and $\mathcal{X} := \{\mathbf{X}_1, \mathbf{X}_2, \ldots, \mathbf{X}_n\}$. The graph-level clustering aims to partition the set $\mathcal{G}$ into a few non-overlapped groups, i.e., $\mathcal{G} = \mathcal{G}^{(1)} \cup \mathcal{G}^{(2)} \cup \cdots \mathcal{G}^{(c)}$ and $\mathcal{G}^{(i)} \cap \mathcal{G}^{(j)} = \emptyset$ for any $i \neq j$, such that the graphs in the same group are similar while the graphs in different groups are dissimilar, without using any label information.

Since the original graph data may not have graph-level feature vectors or they often contain redundant and distracting information, a more effective way is to perform clustering in a latent space given by some representation learning methods. Therefore, we propose to learn latent representations and conduct clustering simultaneously, where the representation learning and clustering facilitate each other. We formalize the objective function for graph-level clustering as follows

$$\mathcal{L}(\phi, \theta) := L_r(g_\phi(\mathcal{X}, \mathcal{G}), \mathcal{X}, \mathcal{G}) + L_{c|\theta}(g_\phi(\mathcal{X}, \mathcal{G})). \tag{1}$$

In (1), $L_r$ denotes the representation learning objective that aims to map the input data $\mathcal{X}, \mathcal{G}$ into a latent space via a deep graph neural network with parameters $\phi$. $L_{c|\theta}$ denotes the clustering objective on the representations $g_\phi(\mathcal{X}, \mathcal{G})$ and is associated with a deep neural network with parameters $\theta$ that may also contain the cluster centers or assignments. Note that there could be a trade-off parameter between $L_r$ and $L_{c|\theta}$, but we just ignore it for convenience. We see that the objective $\mathcal{L}(\phi, \theta)$ does not only learn cluster-oriented representations, but also directly produces clustering results. So there is no need to perform $k$-means or spectral clustering after the pure representation learning like those two-step models mentioned in Section 2.2.

### 3.2 LEARNING GRAPH-LEVEL REPRESENTATIONS

To learn effective representations of the graphs, we take advantages of GNN (Kipf & Welling, 2017; Xu et al., 2019; Wu et al., 2020). GNN leverages the node information and structural information to learn representations for node or graph. GNN aggregates the neighboring information of each node to itself iteratively, thus the learned features could capture both the inherent node information and its neighbors' information. Specifically, the learned feature $\mathbf{h}_v$ for node $v$ in the $k$-th layer can be formulated as follows

$$\mathbf{h}_v^{(k)} = \text{COMBINE}^{(k)} \left( \mathbf{h}_v^{(k-1)}, \mathbf{h}_v^{(k)} \right)$$
$$= \text{COMBINE}^{(k)} \left( \mathbf{h}_v^{(k-1)}, \text{AGGREGATE}^{(k)}(\{\mathbf{h}_u^{(k-1)} : u \in \mathcal{N}(v)\}) \right), \tag{2}$$

where $\mathbf{h}_v^{(k)}$ denotes the aggregated neighbor features in $k$-th layer, $\mathcal{N}(v)$ is the neighborhood set of node $v$. Particularly, the initial representation $\mathbf{h}_v^{(0)}$ is set as the node features of $v$, i.e., $\mathbf{x}_v$. It is worth noting that more global information could be obtained as the layer deepens, while some more generalized information would be possessed in the earlier layers (Xu et al., 2019). Therefore, considering the information from various depths of the network would help us get more powerful representations for graph-level clustering tasks. Following the idea, we concatenate the representation learned at each layer as

$$\mathbf{h}_\phi^i = \text{CONCAT} \left( \{\mathbf{h}_i^{(k)}\}_{k=1}^K \right), \tag{3}$$

where $\mathbf{h}_\phi^i$ is concatenated representation for node $i$, and $\mathbf{h}_i^{(k)}$ is the representation learned in $k$-th layer. After that, we can utilize a READOUT function to obtain the graph-level representation, i.e.,

$$\mathbf{H}_\phi(G_j) = \text{READOUT}(\{\mathbf{h}_\phi^i\}_{i=1}^{|G_j|}), \tag{4}$$

where $|G_j|$ denotes the number of nodes in $G_j$. Therefore, for the given graph dataset $\bar{G} := \{G_j \in \mathcal{G}\}_{j=1}^{n_b}$ in a batch, $\mathbf{H}_\phi(\bar{G}) \in \mathbb{R}^{n_b \times K d_h}$ can be regarded as the learned graph-level representations, where $n_b$ is number of graphs in a batch, $d_h$ is the dimension of each hidden layer of GNN and $K$ is the number of GNN layers. Note that we use the sum readout strategy in this work.

As the graph-level clustering is an unsupervised learning task, it is important to learn more representative features in an unsupervised manner. We follow (Hjelm et al., 2019; Sun et al., 2020) to achieve this by maximizing the mutual information between the representations of entire graphs and substructure, since it has been demonstrated as a powerful unsupervised graph representation learning technique. Specifically, for the given graph datasets in a batch $\bar{G}$ that follows an empirical probability distribution $\mathbb{P}$ on the original data space, the estimator $I_{\phi,\psi}$ of the mutual information (MI) over the global and local pairs is defined as follows:

$$\hat{\phi}, \hat{\psi} = \arg\max_{\phi,\psi} \sum_{\bar{G} \subseteq \mathcal{G}} \frac{1}{|\bar{G}|} \sum_{i \in \bar{G}} I_{\phi,\psi}(\mathbf{h}_\phi^i; \mathbf{H}_\phi(\bar{G})) \triangleq -L_{r|\phi,\psi}, \tag{5}$$

where $|\bar{G}|$ is the number of nodes in $\bar{G}$, $i$ denotes a single node in $\bar{G}$, $I_{\phi,\psi}$ can be parameterized by a discriminator network $T$ with parameter $\psi$. By using Jensen-Shannon MI estimator Nowozin et al. (2016), $I_{\phi,\psi}$ can be formulated as:

$$I_{\phi,\psi}(\mathbf{h}_\phi^i(\bar{G}); \mathbf{H}_\phi(\bar{G})) := \mathbb{E}_{\mathbb{P}}[-\mathrm{sp}(-T_{\phi,\psi}(\mathbf{h}_\phi^i(s); \mathbf{H}_\phi(s)))] - \mathbb{E}_{\mathbb{P} \times \tilde{\mathbb{P}}}[\mathrm{sp}(T_{\phi,\psi}(\mathbf{h}_\phi^i(s'); \mathbf{H}_\phi(s)))], \tag{6}$$

where $s$ denotes the input (positive) sample, and $s'$ denotes the negative sample from the distribution $\tilde{\mathbb{P}}$ that is identical to distribution $\mathbb{P}$. Particularly, the combinations of global (graph-level) and local (node-level) representations in a batch are used to produce negative samples. $\mathrm{sp}(y) = \log(1 + e^y)$ indicates the softplus function. Note that we maximize the MI between graph-level and node-level representations, which facilitates graph-level representations to contain as much information as possible that is shared between node-level representations. It is intuitive that performing $k$-means or spectral clustering directly on the graph-level representations learned seems to be an applicable way, but it often tends to be a trivial solution because the representations learned in this way solely are not guaranteed to be applicable for the graph-level clustering task that we focus in this work.

### 3.3 END-TO-END GRAPH-LEVEL CLUSTERING

To capture more suitable representations for graph-level clustering, we attempt to learn cluster-oriented representations by introducing an explicit clustering objective. Specifically, we propose a clustering network connected with the graph-level features in the representation learning network described above. Then the graph-level features will be projected to the cluster embedding in the low-dimensional latent space, which can be formalized as follows:

$$\mathbf{z}_j = f_\theta(\mathbf{H}_\phi(G_j)), \tag{7}$$

where $\mathbf{z}_j$ denotes the learned cluster embedding for graph $G_j$, and $f_\theta$ is the MLP-based clustering projector with network parameter $\theta$. Let $\mathbf{Z}_{\phi,\theta}(\bar{G}) \in \mathbb{R}^{d_z \times n_b}$ be the cluster embeddings in a batch, where $d_z$ is the dimension of cluster embedding layer. Subsequently, we take inspiration from (Van der Maaten & Hinton, 2008; Xie et al., 2016) to define the graph-level cluster assignment distribution $Q$ based on $\mathbf{Z}_{\phi,\theta}(\bar{G})$ as follows:

$$q_{jt|\phi,\theta} = \frac{(1 + \|\mathbf{z}_j - \boldsymbol{\mu}_t\|^2)^{-1}}{\sum_{t=1}^c (1 + \|\mathbf{z}_j - \boldsymbol{\mu}_t\|^2)^{-1}}, \tag{8}$$

where $\mathbf{z}_j$ is the $j$-th column of $\mathbf{Z}_{\phi,\theta}(\bar{G})$, $c$ is the number of clusters, $\boldsymbol{\mu}_t$ is the $t$-th cluster center that can be initialized by $k$-means, and $q_{jt|\phi,\theta}$ is the graph-level cluster assignment indicating the probability that graph $G_j$ belongs to cluster $t$. Next, we can further define an auxiliary refined cluster assignment distribution $P$ to emphasizes those assignments with high confidence in $Q$ as follows:

$$p_{jt} = \frac{q_{jt|\phi,\theta}^2 / \sum_{j=1}^{n_b} q_{jt|\phi,\theta}}{\sum_{t=1}^c (q_{jt|\phi,\theta}^2 / \sum_{j=1}^{n_b} q_{jt|\phi,\theta})}, \tag{9}$$

where $P$ encourages a more pronounced gap between assignments with high and low probability in $Q$ and can be regarded as pseudo labels for guiding the optimization of $Q$. Therefore, we can define the clustering objective by minimizing the KL-divergence between $P$ and $Q$ as follows:

$$L_{c|\phi,\theta} = KL(P||Q) = \sum_{j=1}^{n_b} \sum_{t=1}^{c} p_{jt} \log \frac{p_{jt}}{q_{jt|\phi,\theta}}. \tag{10}$$

$L_{c|\theta}$ aims to force $Q$ to approximate $P$, i.e., to let $P$ guide the optimization of $Q$ so that the high confident assignment can be emphasized, which can also be regarded as a self-training strategy. By jointly optimizing Eq. 5 and 10, we can construct an end-to-end deep graph-level clustering framework that simultaneously implements graph-level representation learning and clustering. The overall objective of DGLC in terms of minibatch optimization is as follows

$$L_{\text{batch}}(\phi, \psi, \theta) = -\underbrace{\frac{1}{|\bar{G}|} \sum_{i \in \bar{G}} I_{\phi,\psi}(\mathbf{h}_\phi^i; \mathbf{H}_\phi(\bar{G}))}_{L_{r|\phi,\psi}} + \underbrace{\sum_{j=1}^{n_b} \sum_{t=1}^{c} p_{jt} \log \frac{p_{jt}}{q_{jt|\phi,\theta}}}_{L_{c|\phi,\theta}}. \tag{11}$$

## 4 EXPERIMENTS

In this section, we evaluate the proposed method in comparison with several state-of-the-art competitors in graph-level clustering task. We first introduce the datasets and baseline methods used in the experiment and describe the detailed settings of network and parameters. Then, we demonstrate the effectiveness of our method through comprehensive experimental analysis.

### 4.1 DATASET DESCRIPTION AND BASELINE METHODS

**Dataset:** We use six well-known graph datasets in the experiment, including MUTAG[1], PTC-MR[2], PTC-MM[3], BZR[4], ENZYMES[5], COX2[6]. We summarize the information of each dataset in Table 2. More detailed information of each dataset refers to the Appendix A.1.

Table 2: Information of the six benchmark datasets.

| Dataset name | Number of graphs | Range of nodes | Average nodes | Range of edges | Average edges | Classes |
| --- | --- | --- | --- | --- | --- | --- |
| MUTAG | 188 | [10 - 28] | 17.93 | [20 - 66] | 19.79 | 2 |
| PTC-MR | 344 | [2 - 64] | 14.29 | [2 - 142] | 14.69 | 2 |
| PTC-MM | 336 | [2 - 64] | 13.97 | [2 - 142] | 14.32 | 2 |
| BZR | 405 | [13 - 57] | 35.75 | [26 - 120] | 38.36 | 2 |
| ENZYMES | 600 | [2 - 126] | 32.63 | [2 - 298] | 62.14 | 6 |
| COX2 | 467 | [32 - 56] | 41.22 | [68 - 118] | 43.45 | 2 |

**Baseline methods:** We compare our method with six state-of-the-art graph kernel methods including Random walk kernel (RW) (Vishwanathan et al., 2010), Weisfeiler-Lehman kernel (WL) (Shervashidze et al., 2011), Optimal assignment based WL kernel (WL-OA) (Kriege et al., 2016), Shortest path kernel (SP) (Borgwardt & Kriegel, 2005), Lovasz-theta kernel (LT) (Johansson et al., 2014), Graphlet kernel (GK) (Shervashidze et al., 2009), and four unsupervised graph-level representation learning methods including InfoGraph (Sun et al., 2020), Gromov-Wasserstein factorization (GWF) (Xu et al., 2022), Graph contrastive learning (GraphCL) (You et al., 2020), and Joint augmentation optimization (JOAO) (You et al., 2021).

---

[1]https://www.chrsmrrs.com/graphkerneldatasets/MUTAG.zip

[2]https://www.chrsmrrs.com/graphkerneldatasets/PTC-MR.zip

[3]https://www.chrsmrrs.com/graphkerneldatasets/PTC-MM.zip

[4]https://www.chrsmrrs.com/graphkerneldatasets/BZR.zip

[5]http://www.chrsmrrs.com/graphkerneldatasets/ENZYMES.zip

[6]https://www.chrsmrrs.com/graphkerneldatasets/COX2.zip

## 4.2 Experimental settings

For the graph kernel methods we used, they are all normalized with the base graph kernel to be Vertex Histogram kernel if needed, then we directly perform spectral clustering (Ng et al., 2001) on the the similarity matrices produced by them to obtain the clustering results. Note that we also include the $k$-means (Hartigan & Wong, 1979) performance of several graph kernels in the Appendix A.6. While for the unsupervised graph-level representation learning methods, we perform $k$-means (Hartigan & Wong, 1979) and spectral clustering on the learned graph-level representations. Particularly, for GWF (Xu et al., 2022) we not only follow the original paper to perform $k$-means, but also perform spectral clustering to evaluate its clustering performance.

To provide a fair comparison in our experiment, we use exactly the same network architecture as our competitors of unsupervised graph representation learning (Sun et al., 2020; You et al., 2020; 2021), i.e., utilizing the Graph isomorphism network (GIN) (Xu et al., 2019) as the backbone GNN. The cluster projector is constructed with a two-layer MLP-based fully-connected network. We use Adam as the optimizer, the learning rate is chosen from $[10^{-3}, 10^{-5}]$, the batch-size is set to 128 and the total running epoch is set to 20. Moreover, there are three important hyper-parameters in our method, i.e., the layer numbers of GNN, the hidden dimension $d_h$ of each GNN layer and the dimension $d_z$ of the clustering layer. We evaluate the influence of different values of them on the graph-level clustering performance in Appendix A.3 due to the limitation of the paper length.

To evaluate the clustering performance, we consider three popular metrics including clustering accuracy (ACC), normalized mutual information (NMI) and adjusted rand index (ARI). The detailed definition of the three metrics refer to the Appendix A.2. We utilize Pytorch Geometric (Fey & Lenssen, 2019) and GraKeL (Siglidis et al., 2020) libraries to implement our method and other baseline methods. Note that we run all experiments 10 times with NVIDIA Tesla A100 GPU and AMD EPYC 7532 CPU, and report their means and standard deviations.

## 4.3 Experimental results

We compare the proposed DGLC method with 13 baselines and state-of-the-art methods on the six popular benchmarks. The experimental results are shown in Table 3-5, from which we have the following observations.

Table 3: Clustering performance (ACC, NMI, ARI) on MUTAG and PTC-MR. The best result is highlighted in **bold.**

| Method | MUTAG | | | PTC-MR | | |
|---|---|---|---|---|---|---|
| | ACC | NMI | ARI | ACC | NMI | ARI |
| Graph kernel followed by spectral clustering (SC) | | | | | | |
| RW(Vishwanathan et al., 2010)+SC | 77.65±0.00 | 30.81±0.00 | 30.26±0.00 | 56.98±0.00 | 0.63±0.00 | 1.25±0.00 |
| WL(Shervashidze et al., 2011)+SC | 73.40±0.00 | 14.50±0.00 | 21.20±0.00 | 52.91±0.00 | 0.23±0.00 | 0.05±0.00 |
| WL-OA(Kriege et al., 2016)+SC | 67.55±0.00 | 19.64±0.00 | 11.40±0.00 | 59.30±0.00 | 1.77±0.00 | 2.95±0.00 |
| SP(Borgwardt & Kriegel, 2005)+SC | 72.87±0.00 | 10.24±0.00 | 15.95±0.00 | 56.69±0.00 | 1.04±0.00 | 0.50±0.00 |
| LT(Johansson et al., 2014)+SC | 56.60±4.88 | 3.09±1.38 | -0.62±0.63 | 55.17±1.32 | 0.40±0.65 | 0.19±0.52 |
| GK(Shervashidze et al., 2009)+SC | 67.02±0.00 | 1.74±0.00 | 1.04±0.00 | 56.40±0.00 | 1.32±0.00 | 0.31±0.00 |
| Unsupervised graph representation learning followed by $k$-means (KM) and SC | | | | | | |
| InfoGraph(Sun et al., 2020)+KM | 77.95±1.41 | 35.22±3.47 | 30.95±3.03 | 54.79±0.68 | 0.49±0.35 | 0.28±0.21 |
| InfoGraph(Sun et al., 2020)+SC | 72.58±4.83 | 28.68±4.93 | 19.85±5.91 | 56.10±0.33 | 1.50±0.26 | 0.20±0.13 |
| GWF(Xu et al., 2022)+KM | 66.94±7.68 | 12.46±9.31 | 13.32±10.53 | 56.33±3.52 | 1.09±0.88 | 1.65±1.50 |
| GWF(Xu et al., 2022)+SC | 73.92±4.30 | 18.35±3.85 | 24.48±4.69 | 55.32±4.03 | 0.89±0.84 | 1.49±1.44 |
| GraphCL(You et al., 2020)+KM | 77.07±1.21 | 35.69±2.83 | 28.99±2.65 | 54.33±0.76 | 1.15±0.55 | 0.16±0.29 |
| GraphCL(You et al., 2020)+SC | 73.22±2.66 | 32.19±2.05 | 23.44±2.45 | 56.13±0.42 | 1.31±0.30 | 1.17±0.24 |
| JOAO(You et al., 2021)+KM | 79.20±0.72 | **36.32±3.03** | 33.74±1.65 | 56.39±0.18 | 0.53±0.21 | 0.41±0.01 |
| JOAO(You et al., 2021)+SC | 70.72±2.85 | 27.73±0.23 | 17.12±2.03 | 56.16±0.22 | 1.03±0.33 | 0.19±0.11 |
| DGLC(Ours) | **84.68±0.89** | 35.75±2.51 | **47.01±2.64** | **60.93±0.57** | **2.98±0.43** | **4.29±0.52** |

First, graph kernel based graph-level clustering approaches are effective on only few datasets, while achieving mediocre clustering performances on most datasets. For example, RW kernel performs well on MUTAG and PTC-MR, but mediocre on PTC-MM and BZR. While the opposite results

Table 4: Clustering performance (ACC, NMI, ARI) on PTC-MM and BZR. The best result is highlighted in **bold.**

| Method | PTC-MM | | | BZR | | |
|---|---|---|---|---|---|---|
| | ACC | NMI | ARI | ACC | NMI | ARI |
| Graph kernel followed by spectral clustering (SC) | | | | | | |
| RW(Vishwanathan et al., 2010)+SC | 60.71±0.00 | 0.97±0.00 | 2.91±0.00 | 64.69±0.00 | 0.00±0.00 | -0.15±0.00 |
| WL(Shervashidze et al., 2011)+SC | 62.20±0.00 | 1.50±0.00 | 3.87±0.00 | 75.56±0.00 | 0.50±0.00 | 3.76±0.00 |
| WL-OA(Kriege et al., 2016)+SC | 63.39±0.00 | 4.59±0.00 | 2.26±0.00 | 69.63±0.00 | 5.60±0.00 | -8.67±0.00 |
| SP(Borgwardt & Kriegel, 2005)+SC | 62.20±0.00 | 1.63±0.00 | 0.73±0.00 | 79.51±0.00 | 4.13±0.00 | 3.97±0.00 |
| LT(Johansson et al., 2014)+SC | 61.19±0.88 | 0.73±0.55 | 1.09±1.06 | 78.35±0.35 | 0.69±0.28 | 1.12±1.03 |
| GK(Shervashidze et al., 2009)+SC | 62.20±0.00 | 1.63±0.00 | 0.73±0.00 | 61.23±3.36 | 1.06±1.21 | 3.13±3.74 |
| Unsupervised graph representation learning followed by $k$-means (KM) and SC | | | | | | |
| InfoGraph(Sun et al., 2020)+KM | 61.48±1.03 | 2.35±0.83 | 3.61±1.45 | 63.62±2.41 | 1.59±0.95 | 2.39±1.44 |
| InfoGraph(Sun et al., 2020)+SC | 61.96±1.53 | 2.12±0.99 | 4.55±0.83 | 73.53±2.66 | 3.66±2.52 | 5.04±3.12 |
| GWF(Xu et al., 2022)+KM | 53.37±3.18 | 0.30±0.37 | 0.38±1.09 | 53.00±0.31 | 3.42±0.45 | -0.76±0.05 |
| GWF(Xu et al., 2022)+SC | 53.02±1.66 | 0.36±0.28 | 0.21±0.09 | 52.76±0.80 | 3.47±1.16 | -0.71±0.32 |
| GraphCL(You et al., 2020)+KM | 58.93±0.74 | 0.27±0.15 | 0.60±0.14 | 71.43±4.09 | 1.04±0.77 | 3.07±1.03 |
| GraphCL(You et al., 2020)+SC | 62.09±0.56 | 2.14±0.43 | 3.36±0.87 | 72.88±1.66 | 1.90±0.38 | 3.47±0.59 |
| JOAO(You et al., 2021)+KM | 59.04±0.52 | 0.21±0.14 | 0.98±0.41 | 72.64±4.26 | 1.37±1.14 | 4.01±3.39 |
| JOAO(You et al., 2021)+SC | 62.41±0.80 | 2.00±0.78 | 4.28±1.34 | 72.98±1.59 | 2.75±1.30 | 5.62±3.74 |
| DGLC(Ours) | **63.30±0.81** | **2.70±0.45** | **5.53±0.61** | **80.98±0.60** | **9.79±0.92** | **20.53±1.84** |

Table 5: Clustering performance (ACC, NMI, ARI) on ENZYMES and COX2. The best result is highlighted in **bold.**

| Method | ENZYMES | | | COX2 | | |
|---|---|---|---|---|---|---|
| | ACC | NMI | ARI | ACC | NMI | ARI |
| Graph kernel followed by spectral clustering (SC) | | | | | | |
| RW(Vishwanathan et al., 2010)+SC | 17.00±0.00 | 0.66±0.00 | 0.25±0.00 | 51.31±0.00 | 0.70±0.00 | -0.92±0.00 |
| WL(Shervashidze et al., 2011)+SC | 21.00±0.00 | 3.09±0.00 | 1.48±0.00 | 50.54±0.00 | 0.51±0.00 | -0.40±0.00 |
| WL-OA(Kriege et al., 2016)+SC | 20.00±0.00 | 1.35±0.00 | 0.32±0.00 | 50.75±0.00 | 0.51±0.00 | -0.37±0.00 |
| SP(Borgwardt & Kriegel, 2005)+SC | 22.00±0.00 | 2.57±0.00 | 1.69±0.00 | 52.03±0.00 | 0.13±0.00 | 0.01±0.00 |
| LT(Johansson et al., 2014)+SC | 17.00±0.09 | 0.42±0.11 | 0.00±0.00 | 77.52±0.59 | 0.26±0.34 | 0.17±0.71 |
| GK(Shervashidze et al., 2009)+SC | 17.07±0.13 | 0.80±0.25 | 0.00±0.00 | 66.17±0.00 | 0.02±0.00 | 0.08±0.17 |
| Unsupervised graph representation learning followed by $k$-means (KM) and SC | | | | | | |
| InfoGraph(Sun et al., 2020)+KM | 22.06±0.98 | 2.40±0.45 | 1.25±0.52 | 56.74±3.04 | 3.30±0.60 | 0.17±0.10 |
| InfoGraph(Sun et al., 2020)+SC | 23.75±0.50 | 4.64±0.65 | 2.23±0.41 | 70.37±2.01 | **3.56±0.99** | 1.92±1.67 |
| GWF(Xu et al., 2022)+KM | **28.55±0.20** | 6.02±0.55 | **3.16±0.20** | 57.60±4.11 | 1.50±0.13 | 2.08±1.80 |
| GWF(Xu et al., 2022)+SC | 25.66±1.57 | 5.24±1.28 | 1.78±0.61 | 58.83±4.46 | 1.16±0.41 | 1.45±1.21 |
| GraphCL(You et al., 2020)+KM | 21.50±0.22 | 1.55±0.12 | 0.90±0.09 | 68.88±0.59 | 1.05±0.21 | 0.44±0.57 |
| GraphCL(You et al., 2020)+SC | 25.28±0.28 | 4.75±0.36 | 2.03±0.26 | 75.01±2.12 | 1.24±0.37 | 2.39±2.28 |
| JOAO(You et al., 2021)+KM | 21.66±0.37 | 1.60±0.01 | 0.94±0.02 | 70.56±2.03 | 1.19±0.34 | 0.44±0.43 |
| JOAO(You et al., 2021)+SC | 24.65±0.44 | 4.85±0.37 | 2.07±0.18 | 76.46±0.61 | 1.43±0.77 | 2.35±2.49 |
| DGLC(Ours) | 27.08±1.49 | **6.39±1.09** | 2.86±0.80 | **78.28±0.17** | 2.38±0.99 | **6.79±3.37** |

are observed on LT kernel. This is because graph kernels are mainly based on hand-crafted design and are not suitable for arbitrary datasets in practice. Second, the unsupervised graph representation learning methods show potential in handling graph-level clustering. For example, JOAO obtains encouraging performance on MUTAG, BZR and COX2. GWF achieves state-of-the-art performance on ENZYMES. Although such methods achieve promising graph-level clustering performance in many cases, they still suffer from the undesirable graph-level representations learned for clustering, i.e., their representation learning do not explicitly optimize for the clustering task. Third, the proposed DGLC method outperforms both types of the above solutions with a large margin in most cases. For example, DGLC outperforms the runner-up with 5.48% and 13.27% advantages on MUTAG in terms of ACC and ARI, and with 1.47%, 5.66% and 14.91% advantages on BZR in terms of ACC, NMI, and ARI. This fully demonstrates the effectiveness of our method. Compared with

graph kernel based approaches, DGLC is more general for different types of graph data. Compared with the latest unsupervised graph representation learning approaches, DGLC has a clear clustering objective in the optimization and thus tends to learn clustering-oriented graph-level representations and achieves state-of-the-art performance.

## 4.4 QUALITATIVE STUDY

In this section, we conduct a qualitative study to provide visual comparison for the graph-level clustering. Specifically, we compare our method with several state-of-the-art unsupervised graph representation learning methods including InfoGraph, GWF, GrahCL and JOAO by utilizing t-SNE (Van der Maaten & Hinton, 2008) and visualize their learned graph-level representations on MUTAG and ENZYMES. The visualization results are shown in Figure 1. We can observe that

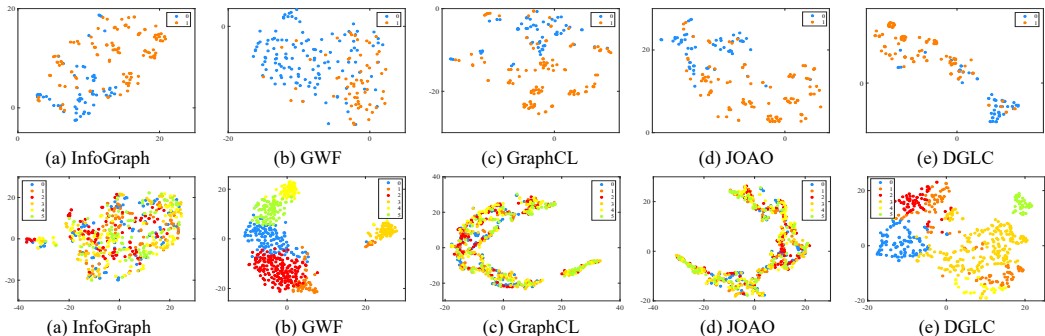

Figure 1: t-SNE visualization of the learned graph-level representations of our methods and other unsupervised graph representation learning methods. The first row is the visualization for MUTAG, while the second row is for ENZYMES.

compared with other methods, DGLC explicitly reveals more compact intra-class structure and more distinct inter-class discrepancy. For example, the learned representations of the two classes in MU-TAG are more separated in our method compared to others. Besides, we can find that InfoGraph, GraphCL and JOAO fail to capture good clustering structure for ENZYMES, while GWF and ours do. In general, the visualization results of the learned graph-level representations also support the effectiveness of our method.

## 4.5 PARAMETER SENSITIVITY ANALYSIS AND ABLATION STUDY

To evaluate the robustness of DGLC and the effectiveness of each component, we conduct the parameter sensitivity analysis and ablation study. Please see Appendix A.3 and A.4 for the experimental results and discussions due to the limitation of the paper length.

## 4.6 COMPUTATIONAL TIME COMPARISON

We also demonstrate the time efficiency of DGLC by comparing the running time with several graph kernels and unsupervised graph representation learning baselines. Please see Appendix A.5 for the experimental results and discussions due to the limitation of the paper length.

## 5 CONCLUSION

This work has studied the problem of graph-level clustering and proposed an end-to-end deep graph-level clustering method based on deep graph neural network. The proposed DGLC method leverages the powerful representation learning capability of GIN and defines an explicit clustering objective to help learn cluster-favor representations for graph-level clustering. We compared the proposed method with two types of baselines, one is based on graph kernels followed by spectral clustering and the other is based on graph-level representation learning followed by $k$-means and spectral clustering. The experiments on six graph datasets have showed that our method has much higher clustering accuracy than the baselines.

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

## A   APPENDIX

### A.1   DETAILED INFORMATION OF DATASET

We provide the detailed information of six graph datasets used in our experiment here:

- **MUTAG** is a compound dataset that contains 188 compounds, which are grouped into 2 categories based on the mutagenic effect of them to a bacterium. Note molecules possess natural graph structure, where they are expressed by average 17.93 nodes (for atoms) and 19.79 edges (for chemical bonds).

- **PTC-MR** and **PTC-MM** are the subset of PTC dataset, which is a compound dataset that divided into 2 categories based on the carcinogenicity to rodents. Note that PTC-MR contains 344 compounds with average 14.29 nodes and 14.69 edges, while PTC-MM contains 336 compounds with average 13.97 ndoes and 14.32 edges, respectively.

- **BZR** is the ligand dataset for benzodiazepine receptor, which are divided into 2 classes according to the activity and inactivity of compounds. Note that BZR contains 405 graphs in total with average 35.75 nodes and 38.36 edges per graph.

- **ENZYMES** contains 600 protein data for 6 classes of enzymes, with 100 proteins per class. Each protein data can be represented as a graph with average 32.63 nodes and 62.14 edges.

- **COX2** consists of 467 inhibitor for cyclooxygenase-2 and are divided into 2 classes based on whether the compounds are active or inactive. Note that each graph in this dataset is with average 41.22 nodes and 43.45 edges.

### A.2   DEFINITION OF THREE CLUSTERING METRICS

In this section, we introduce three clustering metrics used in this paper, with $y_j$ and $\hat{y}_j$ denoting the true labels and the predicted labels for graph $G_j$ respectively.

**Clustering accuracy (ACC):**   ACC is expressed as the comparison of the true labels and predicted labels leveraged on sample size $n$, which is defined as follows:

$$\text{ACC} = \frac{\sum_{i=1}^{n} \delta\left(y_j, \hat{y}_j\right)}{n}, \text{ where } \delta(x, y) = \left\{ \begin{array}{ll} 1 & \text{if } x = y \\ 0 & \text{otherwise} \end{array} \right\} \tag{12}$$

**Normalized mutual information (NMI):**   NMI score scales the mutual information scores by some generalized mean of entropy of true label set $\Omega$ and cluster label set $C$. It can be formalized as follows:

$$\text{NMI}(\Omega, C) = \frac{I(\Omega; C)}{(H(\Omega) + H(C))/2} \tag{13}$$

where $I(\Omega; C) = H(\Omega) + H(C) - H(\Omega, C)$ denotes the mutual information between $\Omega$ and $C$, and $H(\cdot)$ is the information entropy.

**Adjusted rand index (ARI):**   ARI score is an adjusted score of Rand index (RI) for chance. RI is also a similarity measure by considering all pairs of samples and counting pairs that are assigned in

the same or different clusters in the predicted and true labels. ARI can be formalized as follows:

$$
\begin{aligned}
\text{ARI} &= \frac{(\text{RI} - \text{Expected RI})}{(\max(\text{RI}) - \text{Expected RI})} \\
&= \frac{\sum_{ij} \binom{n_{ij}}{2} - \left[ \sum_i \binom{a_i}{2} \sum_j \binom{b_j}{2} \right] / \binom{n}{2}}{\frac{1}{2} \left[ \sum_i \binom{a_i}{2} + \sum_j \binom{b_j}{2} \right] - \left[ \sum_i \binom{a_i}{2} \sum_j \binom{b_j}{2} \right] / \binom{n}{2}}
\end{aligned}
\tag{14}
$$

where $a_i = \sum_{j=1}^r n_{ij}$, $b_i = \sum_{i=1}^s n_{ij}$, $n_{ij}$ denotes an entry from the contingency table of cluster $i$ and class $j$, $r$ and $s$ are numbers of clusters and classes.

Note that ACC and NMI range from $[0, 1]$, while ARI ranges from $[-1, 1]$. The higher values of ACC, NMI and ARI represent the better clustering performance.

### A.3 PARAMETER SENSITIVITY ANALYSIS

We analyze the sensitivity of DGLC to the hyperparameters, i.e., the hidden dimension $d_h$ of GNN layers, the embedding dimension $d_z$ of clustering layer and the number of GNN layers. Here we take MUTAG and PTC-MR datasets as the example to evaluate the influence of the change of $d_h$ and $d_z$ values. Specifically, we select the values of $d_h$ in $[16, 32, \ldots, 256]$ and $d_z$ in $[5, 10, \ldots, 30]$, the results are shown in Figure 2. We can observe that the accuracy on both datasets are relatively stable, showing little fluctuation when parameters vary. In contrast, NMI and ARI are of high performance when the selection of parameters are moderate. In general, DGLC shows robust performance against the two parameters. Nevertheless, we recommend to choose $d_z$ from 10 to 25 and $d_h$ from 32 to 128 to obtain better clustering performance in practice. Except for the ones mentioned above, we further conduct the sensitivity analysis on the number of GNN hidden layers on three datasets (MUTAG, PTC-MR and BZR). We vary the number of GNN hidden layers in $[2, 3, \ldots, 10]$. The experimental results are shown in Figure 3. It could be seen that PTC-MR is quite stable for all three metrics. For MUTAG and BZR, whereas, DGLC shows better performance when setting the number of GNN hidden layers to 4 and 5. In general, DGLC obtains relatively stable performance at different numbers of GNN layers, despite fluctuations at some specific fetch values.

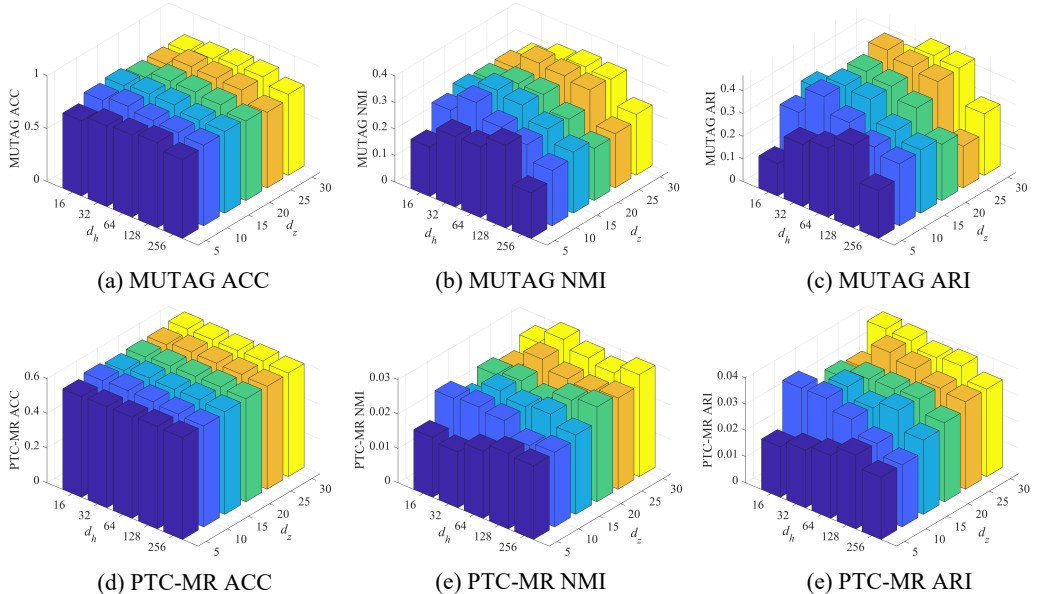

|   |   |   |
|---|---|---|
| (a) MUTAG ACC | (b) MUTAG NMI | (c) MUTAG ARI |
| (d) PTC-MR ACC | (e) PTC-MR NMI | (e) PTC-MR ARI |

Figure 2: Sensitivity analysis of accuracy, NMI and ARI regarding the dimension $d_h$ of GNN hidden layers and the embedding dimension $d_z$ of clustering layer on MUTAG and PTC-MR datasets.

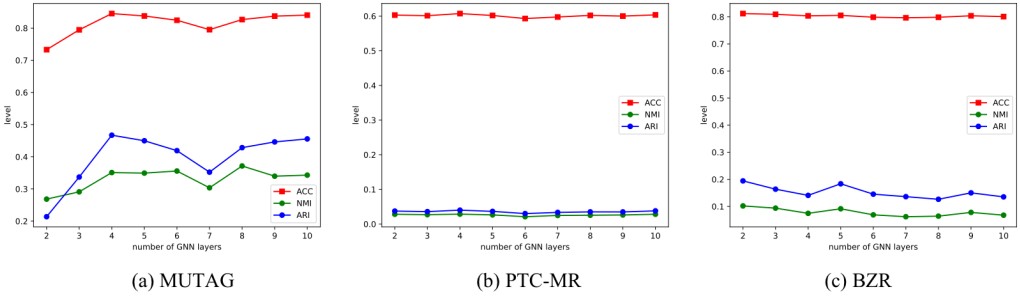

(a) MUTAG        (b) PTC-MR        (c) BZR

Figure 3: Sensitivity analysis of ACC, NMI and ARI regarding the number of GNN hidden layers on MUTAG, PTC-MR and BZR datasets

## A.4 ABLATION STUDY

In this section, we conduct experiments to evaluate the influence of each proposed strategy on our method. Specifically, we construct four degradation models of our method by respectively removing some components of it. There are:

- $\text{DGLC}_{d1}$: We remove the clustering loss and joint training strategy of DGLC and evaluate the model by performing $k$-means on the learned graph-level representations, i.e., the model can be regarded as InfoGraph in this way.

- $\text{DGLC}_{d2}$: We keep the clustering loss and joint training strategy while directly using $k$-means to produce the clustering results instead of producing the clustering labels with the cluster label assignment $Q$.

- $\text{DGLC}_{d3}$: We degrade DGLC as a two-stage model, i.e., we train the model by respectively optimizing the graph representation learning objective and clustering objective. The clustering results are still obtained from the graph-level cluster assignment $Q$ in the second training stage.

Table 6: Clustering performance (ACC, NMI, ARI) on MUTAG and BZR. The best result is highlighted in bold.

| Method | MUTAG | | | BZR | | |
|---|---|---|---|---|---|---|
| | ACC | NMI | ARI | ACC | NMI | ARI |
| $\text{DGLC}_{d1}$ | 77.95±1.41 | 35.22±3.47 | 30.95±3.03 | 63.62±2.41 | 1.59±0.95 | 2.39±1.44 |
| $\text{DGLC}_{d2}$ | 80.50±2.34 | 32.52±3.65 | 37.16±5.53 | 66.61±3.14 | 1.98±1.29 | 4.32±2.54 |
| $\text{DGLC}_{d3}$ | 81.48±2.31 | 30.89±3.98 | 38.34±6.61 | 73.87±2.58 | 2.92±2.30 | 5.35±3.96 |
| DGLC | **84.68±0.89** | **35.75±2.51** | **47.01±2.64** | **80.98±0.60** | **9.79±0.92** | **20.53±1.84** |

We run experiments on MUTAG and BZR to evaluate their performance. Table 6 summarizes the experimental results, from which we have the following observations:

- Both $\text{DGLC}_{d2}$ and $\text{DGLC}_{d3}$ significantly outperform $\text{DGLC}_{d1}$, which fully suggests that learning clustering-oriented representations would benefit graph-level clustering.

- Producing clustering results from the graph-level cluster assignment $Q$ is more reasonable as the clustering performance degrades when directly performing $k$-means on the learned cluster embeddings.

- Joint training with representation learning and clustering objectives yields better clustering performance. For example, DGLC outperforms $\text{DGLC}_{d3}$ by 3.20%, 4.86%, 8.67% in terms of ACC, NMI and ARI on MUTAG.

## A.5 Computational time comparison

In this section, we compare the proposed DGLC with some baseline methods to demonstrate its efficiency in time consuming. Specifically, for graph kernels, we select RW(Vishwanathan et al., 2010), WL(Shervashidze et al., 2011), SP(Borgwardt & Kriegel, 2005) and LT(Johansson et al., 2014) as our competitors. For unsupervised graph representation learning methods, we select GWF (Xu et al., 2022) and InfoGraph (Sun et al., 2020). Note that we run 20 epochs for GWF, InfoGraph and DGLC for fair comparison. Table 7 shows the running times of each method on six benchmark datasets used in this paper. We can see that RW, LT and GWF are quite time consuming, especially on datasets like ENZYMES and COX2 that contain numerous nodes and edges. In contrast, WL, SP, InfoGraph and DGLC are much more efficient compared with them and have comparable time efficiency.

Table 7: Running time comparison (in seconds) on the six benchmark graph datasets.

| Method | MUTAG | PTC-MR | BZR | PTC-MM | ENZYMES | COX2 |
|---|---|---|---|---|---|---|
| RW(Vishwanathan et al., 2010)+SC | 12.29 | 29.29 | 76.34 | 25.44 | 2346.51 | 2457.56 |
| WL(Shervashidze et al., 2011)+SC | 2.19 | 4.97 | 9.57 | 7.15 | 13.43 | 10.65 |
| SP(Borgwardt & Kriegel, 2005)+SC | 3.60 | 5.38 | 25.49 | 5.08 | 53.75 | 32.39 |
| LT(Johansson et al., 2014)+SC | 88.28 | 160.86 | 860.70 | 552.66 | 9117.17 | 6016.26 |
| InfoGraph(Sun et al., 2020)+KM | 9.23 | 10.96 | 23.42 | 11.48 | 29.37 | 28.99 |
| InfoGraph(Sun et al., 2020)+SC | 35.96 | 96.60 | 165.48 | 101.2 | 313.70 | 300.84 |
| GWF(Xu et al., 2022)+KM | 477.48 | 830.26 | 2480.76 | 803.81 | 3668.92 | 2945.12 |
| GWF(Xu et al., 2022)+SC | 566.41 | 911.37 | 2591.73 | 896.44 | 3954.87 | 3132.67 |
| DGLC | 10.16 | 12.12 | 25.66 | 12.84 | 31.87 | 30.50 |

## A.6 $k$-means performance of Graph kernels

Here we provide $k$-means performance of some Graph kernels, including, RW(Vishwanathan et al., 2010), WL(Shervashidze et al., 2011), WL-OA(Kriege et al., 2016), and SP(Borgwardt & Kriegel, 2005). The experimental results are shown in Table 8, 9, 10, and the proposed DGLC method outperforms them significantly.

Table 8: Clustering performance (ACC, NMI, ARI) on MUTAG and PTC-MR. The best result is highlighted in **bold.**

| Method | MUTAG | | | PTC-MR | | |
|---|---|---|---|---|---|---|
| | ACC | NMI | ARI | ACC | NMI | ARI |
| Graph kernel followed by $k$-means (KM) | | | | | | |
| RW(Vishwanathan et al., 2010)+KM | 77.66±0.00 | 30.82±0.00 | 30.26±0.00 | 51.16±0.00 | 0.19±0.00 | -0.55±0.00 |
| WL(Shervashidze et al., 2011)+KM | 73.94±0.00 | 15.51±0.00 | 22.25±0.00 | 57.56±0.00 | 1.10±0.00 | 1.89±0.00 |
| WL-OA(Kriege et al., 2016)+KM | 73.94±0.00 | 16.92±0.00 | 22.42±0.00 | 55.81±0.00 | 0.59±0.00 | 0.99±0.00 |
| SP(Borgwardt & Kriegel, 2005)+KM | 76.06±0.00 | 15.38±0.00 | 25.11±0.00 | 59.30±0.00 | 1.87±0.00 | 2.73±0.00 |
| DGLC(Ours) | **84.68±0.89** | **35.75±2.51** | **47.01±2.64** | **60.93±0.57** | **2.98±0.43** | **4.29±0.52** |

## A.7 Experiment on large-scale dataset

To validate the effectiveness of the proposed method on large-scale graph datasets, we supplement two more datasets in our experiment. Specifically, we choose NCI1, NCI109, and COLLAB datasets to conduct experiment, the detail information of the three datasets are shown in Table 11. The experiment results are shown in Table 12 and Table 13. We can see that almost graph kernels show low efficiency and bad clustering performance when handling large-scale datasets, some of them are too time consuming. While the proposed DGLC method shows superiority compared with graph kernels and graph representation learning methods. DGLC obtain the best clustering performance in most cases. Besides, the experiment on COLLAB, which contains 3 classes, also demonstrates the effectiveness of the proposed DGLC method in processing datasets containing more than 2 classes.

Table 9: Clustering performance (ACC, NMI, ARI) on PTC-MM and BZR. The best result is highlighted in **bold.**

| Method | PTC-MM | | | BZR | | |
|---|---|---|---|---|---|---|
| | ACC | NMI | ARI | ACC | NMI | ARI |
| Graph kernel followed by $k$-means (KM) | | | | | | |
| RW(Vishwanathan et al., 2010)+KM | 55.06±0.00 | 0.02±0.00 | 0.00±0.00 | 58.52±0.00 | 0.19±0.00 | -1.55±0.00 |
| WL(Shervashidze et al., 2011)+KM | 58.63±0.00 | 0.82±0.00 | 2.15±0.00 | 68.15±0.00 | 0.98±0.00 | 5.17±0.00 |
| WL-OA(Kriege et al., 2016)+KM | 58.04±0.00 | 0.81±0.00 | 1.93±0.00 | 67.90±0.00 | 2.17±0.00 | -6.78±0.00 |
| SP(Borgwardt & Kriegel, 2005)+KM | 61.01±0.00 | 0.85±0.00 | 2.67±0.00 | 65.43±0.00 | 0.27±0.00 | 2.36±0.00 |
| DGLC(Ours) | **63.30±0.81** | **2.70±0.45** | **5.53±0.61** | **80.98±0.60** | **9.79±0.92** | **20.53±1.84** |

Table 10: Clustering performance (ACC, NMI, ARI) on ENZYMES and COX2. The best result is highlighted in **bold.**

| Method | ENZYMES | | | COX2 | | |
|---|---|---|---|---|---|---|
| | ACC | NMI | ARI | ACC | NMI | ARI |
| Graph kernel followed by $k$-means (KM) | | | | | | |
| RW(Vishwanathan et al., 2010)+KM | 23.17±0.00 | 2.50±0.00 | 1.74±0.00 | 53.96±0.00 | 0.60±0.00 | -1.68±0.00 |
| WL(Shervashidze et al., 2011)+KM | 21.50±0.00 | 2.18±0.00 | 0.96±0.00 | 50.96±0.00 | 0.54±0.00 | -0.33±0.00 |
| WL-OA(Kriege et al., 2016)+KM | 20.83±0.00 | 1.68±0.00 | 0.55±0.00 | 50.75±0.00 | 0.51±0.00 | -0.37±0.00 |
| SP(Borgwardt & Kriegel, 2005)+KM | 22.17±0.00 | 2.79±0.00 | 1.70±0.00 | 52.03±0.00 | 0.13±0.00 | 0.01±0.00 |
| DGLC(Ours) | **27.08±1.49** | **6.39±1.09** | **2.86±0.80** | **78.28±0.17** | **2.38±0.99** | **6.79±3.37** |

Table 11: Information of the three large-scale datasets.

| Dataset name | Number of graphs | Average nodes | Average edges | Classes |
|---|---|---|---|---|
| NCI1 | 4,110 | 29.87 | 32.30 | 2 |
| NCI109 | 4,127 | 14.29 | 14.69 | 2 |
| COLLAB | 5,000 | 74.49 | 2457.78 | 3 |

Table 12: Clustering performance (ACC, NMI, ARI) on NCI1 and NCI109. The best result is highlighted in **bold**. N/A denotes the results are unavailable (out of memory or the running time over 24 hours).

| Method | NCI1 | | | NCI109 | | |
|---|---|---|---|---|---|---|
| | ACC | NMI | ARI | ACC | NMI | ARI |
| Graph kernel followed by spectral clustering (SC) | | | | | | |
| RW(Vishwanathan et al., 2010)+SC | N/A | N/A | N/A | N/A | N/A | N/A |
| WL(Shervashidze et al., 2011)+SC | 50.05±0.00 | 0.00±0.00 | 0.00±0.00 | 50.39±0.00 | 0.01±0.00 | 0.00±0.00 |
| SP(Borgwardt & Kriegel, 2005)+SC | 50.10±0.00 | 0.10±0.00 | 0.00±0.00 | 52.26±0.00 | 0.33±0.00 | 0.19±0.00 |
| LT(Johansson et al., 2014)+SC | N/A | N/A | N/A | 50.47±0.29 | 0.01±0.01 | -0.01±0.01 |
| Unsupervised graph representation learning followed by $k$-means (KM) and SC | | | | | | |
| InfoGraph(Sun et al., 2020)+KM | 54.11±2.15 | 1.28±1.11 | 0.85±0.87 | 54.38±1.85 | 1.25±0.74 | 0.89±0.68 |
| InfoGraph(Sun et al., 2020)+SC | 54.87±1.68 | 0.93±0.56 | 1.04±0.76 | 54.67±1.93 | 1.08±0.52 | 1.00±0.80 |
| GraphCL(You et al., 2020)+KM | 55.37±1.66 | 0.47±0.28 | 0.99±0.82 | 55.37±1.68 | 1.79±0.93 | 2.11±1.44 |
| GraphCL(You et al., 2020)+SC | 55.93±1.24 | 0.61±0.63 | 1.08±0.79 | 56.29±2.24 | 2.12±1.16 | **2.48±2.79** |
| JOAO(You et al., 2021)+KM | 51.12±0.37 | 0.43±0.18 | 0.05±0.03 | 56.20±0.58 | 1.73±0.72 | 1.54±0.28 |
| JOAO(You et al., 2021)+SC | 51.48±2.98 | 0.88±1.22 | 0.40±1.17 | 56.30±0.85 | **4.61±0.43** | 1.81±0.44 |
| DGLC(Ours) | **57.69±2.31** | **2.50±0.89** | **2.56±1.39** | **56.36±2.31** | 1.94±0.89 | 1.81±1.38 |

Table 13: Clustering performance (ACC, NMI, ARI) on COLLAB. The best result is highlighted in **bold**. N/A denotes the results are unavailable (out of memory or the running time over 24 hours).

| Method | COLLAB | | |
| --- | --- | --- | --- |
| | ACC | NMI | ARI |
| Graph kernel followed by spectral clustering (SC) | | | |
| RW(Vishwanathan et al., 2010)+SC | N/A | N/A | N/A |
| WL(Shervashidze et al., 2011)+SC | 53.20±0.00 | 1.96±0.00 | 0.53±0.00 |
| SP(Borgwardt & Kriegel, 2005)+SC | 48.72±0.00 | 17.91±0.00 | **13.93±0.00** |
| LT(Johansson et al., 2014)+SC | N/A | N/A | N/A |
| Unsupervised graph representation learning followed by $k$-means (KM) and SC | | | |
| InfoGraph(Sun et al., 2020)+KM | 59.64±1.78 | 14.40±2.93 | 6.61±2.27 |
| InfoGraph(Sun et al., 2020)+SC | 60.92±2.49 | 15.37±3.28 | 9.33±3.45 |
| GraphCL(You et al., 2020)+KM | 58.02±1.22 | 17.81±1.94 | 11.33±0.56 |
| GraphCL(You et al., 2020)+SC | 57.83±0.61 | 16.97±1.25 | 10.10±0.65 |
| JOAO(You et al., 2021)+KM | 58.34±1.46 | 18.73±2.62 | 11.06±1.79 |
| JOAO(You et al., 2021)+SC | 57.84±0.88 | 17.12±2.13 | 10.55±0.84 |
| DGLC(Ours) | **61.15±1.44** | **19.98±1.41** | 12.17±2.03 |

