# OpenReview forum: "Deep Graph-Level Clustering Using Pseudo-Label-Guided Mutual Information Maximization Network"
_ICLR.cc/2023/Conference — Submitted to ICLR 2023_

### Official Review · Reviewer_jD5G · 2022-10-21

**Confidence:** 4
**Correctness:** 2
**Technical Novelty And Significance:** 1
**Empirical Novelty And Significance:** 1
**Recommendation:** 3

**Clarity, Quality, Novelty And Reproducibility:**

The quality of the work is poor because of a lack of innovation in motivation and method. The idea is relatively clear, the two modules of graph representation and clustering are combined to optimize each other to form an end-to-end framework, the originality is poor, the graph representation method and the clustering method are all previous work, and the changes are not large, so I think the originality is lacking.

**Strength And Weaknesses:**

Strength: An end-to-end method is proposed to jointly optimize the two modules of graph representation and clustering.
Weaknesses:
(1) The motivation of this paper is that the similarity of the graph level is difficult to measure, which is a very common question that most people have asked. The proposed measurement method is not novel, and the only thing is to reduce the dimensionality of the feature.
(2) As far as I know, there are many works on graph clustering before, which are not mentioned in the related works of this article.
(3) The graph representation module of the proposed method is the previous work, and the clustering module is also the previous DEC clustering method. It's more like putting together previous work, which is not innovative enough.
(4) In the experimental part, there is no graph-level clustering work for comparison. Some of the methods compared in this paper are graph representation and k-means or SC, so whether other graph representation methods plus the clustering method proposed in this paper (that is, DEC) are better than the effect of this paper.

**Summary Of The Paper:**

In this paper, the authors aim to solve the measurement problem of graph-level clustering, and proposes an end-to-end method to jointly optimize graph representation and clustering, and the clustering goal can guide the learning of the entire graph, which is more effective than two stages.

**Summary Of The Review:**

The motivation for this paper is that graph-level similarity is difficult to measure, and the solution is to map to a low-dimensional space (represented by a graph) and then measure similarity through an MLP layer, which lacks innovation. In addition, the proposed graph clustering method lacks novelty, and the related work does not summarize the previous graph clustering work, and the experiment is relatively insufficient.

---

### Official Review · Reviewer_RqLm · 2022-10-23

**Confidence:** 4
**Correctness:** 3
**Technical Novelty And Significance:** 2
**Empirical Novelty And Significance:** 2
**Recommendation:** 5

**Clarity, Quality, Novelty And Reproducibility:**

The novelty of this paper is limited as the proposed method simply combines DEC and InfoGraph without any modification in the graph-level clustering problem. Despite some grammar mistakes and typos, the paper is easy to follow and thus the overall presentation of this paper is good. As for reproducibility, the data sets and code are publicly available.

**Strength And Weaknesses:**

Strength:
1.	The experimental results show the effectiveness of the proposed method over SOTA methods.
2.	The paper is well-organized. The presentation of this paper is good, and most parts of the paper are easy to follow.
3.	The datasets are publicly available, and the proposed method is easy to implement.
Weakness:
1.	The novelty of this paper is limited. The proposed method simply combines DEC[1] and InfoGraph [2].
2.	Some grammar mistakes and typos should be corrected, such as in the first paragraph on page 6, “which is also can also be regarded as a self-training strategy” and the missing conjunction ‘and’ in the sentence below equation 2.
3.	The motivation of the proposed method is not strong enough.


**Summary Of The Paper:**

This paper mainly tackles the graph-level clustering problem by proposing a novel method called Deep Graph-Level Clustering (DGLC) to learn graph-level representations and maximize the mutual information between the representations of entire graphs and substructures. Specifically, a clustering objective is proposed to guide the representation learning for the graph via pseudo labels. In addition, the authors analyze the effectiveness of graph kernel-based methods and unsupervised graph representation learning methods.

**Summary Of The Review:**

This paper mainly tackles the graph-level clustering problem by proposing a novel method called Deep Graph-Level Clustering (DGLC). Despite a few grammar mistakes and typos, the presentation of this paper is good as most parts of the paper are easy to follow. The novelty of this paper is limited. The proposed method simply combines DEC[1] and InfoGraph [2]. In addition, the motivation of the proposed method is not strong enough. The authors mainly focus on discussing the challenges of the graph kernel-based methods and unsupervised learning-based methods on graph-level clustering problems, while the other existing methods, such as InfoGraph, have already addressed most of these challenges (it’s difficult and ineffective to measure the similarity between graphs for graph kernel-based methods and some existing methods do not utilize the label information to better represent graphs in latent space). In the experiment, the results show the effectiveness of the proposed method over SOTA methods, and the data sets and the code are publicly available.
In the experiment, the authors categorize InfoGraph as an unsupervised method, but InfoGraph consists of a supervised loss term considering the label information. It is a little bit unfair to measure the performance of InfoGraph in the unsupervised setting by simply removing the supervised loss term. In addition, the proposed method is similar to InfoGraph, and the main difference is that DGLC benefits from the joint training strategy, while InfoGraph doesn’t. In Table 7, the authors show the running time of the proposed method and SOTA methods. Why does InfoGraph+KM take more time than DGLC? DGLC and InfoGraph both share the same graph-level representation learning strategy, while InfoGraph does not have the clustering module during the training (if KM or SC is only used after the training stage is over) and DGLC consists of a clustering module at the training stage. It seems that the running time of InfoGraph should be smaller than DGLC.
In the experiment, the numbers of graphs in all data sets are too small. The authors should experiment on some large graph data sets, such as RDT-M5K data set or QM9 data set.
One minor question: How do you get the negative sample s' in equation 6?


[1] Junyuan Xie, Ross Girshick, and Ali Farhadi. Unsupervised deep embedding for clustering analysis. In Proceedings of the International Conference on Machine Learning, pp. 478–487. PMLR, 2016.
[2] Fan-Yun Sun, Jordon Hoffman, Vikas Verma, and Jian Tang. Infograph: Unsupervised and semisupervised graph-level representation learning via mutual information maximization. In Proceedings of the International Conference on Learning Representations, 2020.

---

### Official Review · Reviewer_vbKc · 2022-10-25

**Confidence:** 4
**Correctness:** 3
**Technical Novelty And Significance:** 3
**Empirical Novelty And Significance:** 2
**Recommendation:** 6

**Clarity, Quality, Novelty And Reproducibility:**

The paper is clearly written. Some steps in the theoretical description could be better elucidated, although the paper is overall well structured. Experiments are well presented to allow for reproducibility.

**Strength And Weaknesses:**

The method is well supported by a solid theoretical foundation. While not being an entirely novel idea, the method efficiently combines different elements resulting in an innovative approach. I think the experiments could be more comprehensive. Some general comments are given below.

- The authors could better elucidate the contribution of the node label type in their approach. While they mention the label information is not used in the clustering, it would be useful to compare the impact of categorical vs continuous node labels. This is relevant both on the theoretical and experimental side. From the experimental perspective, they chose kernels which cannot handle the continuous node labels, making the comparison not fully fair with the deep learning approaches.

- Only using datasets with 2 classes is quite limiting. Those data are typically used for classification, and I believe 2 classes are not enough to assess the potential of clustering based methods, since the granularity and differences among the classes itself are limited

- Some benchmark datasets as well as methods are missing. On the graph kernels side, NCI1 and NCI109 datasets should a least be included. Furthermore, optimal assignment based kernels could also be integrated in the comparison. Additionally, many graph kernels also provide a feature representation, therefore one could also use k-means on these  for a fair comparison.


**Summary Of The Paper:**

The paper proposes a new end-to-end deep learning method for graph clustering. The goal is to partition a set of graphs according to the similarities of their structures, while learning a graph-level representations. Mutual information theory is exploited in order to maximize within cluster similarities, while penalizing the between clusters similarity

**Summary Of The Review:**

Overall, I don't see major flows in the paper. The proposed method brings an innovative component, while the experiments could benefit from improvements.

---

### Decision · Program_Chairs · 2023-01-20

**Decision:**

Reject

**Justification For Why Not Higher Score:**

The novelty of the paper is limited and the problem is not of interest for a broad audience.

**Justification For Why Not Lower Score:**

N/A

**Metareview: Summary, Strengths And Weaknesses:**

The authors present a new algorithm for graph clustering. In this problem a set of graphs is given in input and the output is formed by a partitioning of the input graph in clusters of similar graphs.  To solve this problem the authors present Deep Graph-Level Clustering (DGLC), an end-to-end system that learns representations and clustering of the graphs at the same time.

The paper contains some interesting ideas and nice experimental results although it also has some important limitations.

First, the novelty of the introduced method is limited. In fact, the method is based on a careful combination of two existing methods DEC and InfoGraph.

Second, the motivations behind the problem are also not too convincing. It would be good for the authors to provide more examples of applications.

Overall, the paper is interesting but it does not meet the ICLR acceptance bar.